# Effect of Choline-Based Deep Eutectic Solvent Pretreatment on the Structure of Cellulose and Lignin in Bagasse

Cuicui Li, Chongxing Huang *, Yuan Zhao, Chaojian Zheng, Hongxia Su, Lanyu Zhang, Wanru Luo, Hui Zhao, Shuangfei Wang and Li-Jie Huang

College of Light Industry and Food Engineering, Guangxi University, Nanning 530004, China; licuicui081@163.com (C.L.); zy199113@163.com (Y.Z.); z2722322150@163.com (C.Z.); suhongxia@st.gxu.edu.cn (H.S.); zhanglanyu20202021@163.com (L.Z.); raremiio@163.com (W.L.); zhh@gxu.edu.cn (H.Z.); wangsf@gxu.edu.cn (S.W.); jiely165@163.com (L.-J.H.)
* Correspondence: huangcx21@163.com

**Abstract:** Deep eutectic solvents (DESs) is a newly developed green solvent with low cost, easy preparation and regeneration. Because of its excellent solubility and swelling effect in lignocellulose, it has received widespread attention and recognition. In this study, choline-based deep eutectic solvents (DESs)—choline chloride-urea (CC-U), choline chloride-ethylene glycol (CC-EG), choline chloride-glycerol (CC-G), choline chloride-lactic acid (CC-LA), and choline chloride-oxalic acid (CC-OA)—were used to extract and separate bagasse. The effects of hydrogen bond donors on lignin separation and the fiber and lignin structure were investigated. All five DESs could dissolve lignin from bagasse; acidic DESs exhibited higher solubility than basic DESs. CC-OA effectively separated lignin and hemicellulose. CC-LA showed weaker lignin separation ability than CC-OA. CC-G, CC-EG, and CC-U were more inclined to selectively separate lignin than hemicellulose. The crystalline cellulose II structure was retained after DES pretreatment. Acidic DESs effectively improved the crystallinity of bagasse fiber; the crystallinities for CC-OA and CC-LA pretreatment were 62.26% and 61.65%, respectively. The lignin dissolved in DES was mainly syringyl lignin. The lignin dissolved in CC-U, CC-LA, and CC-OA contained a small amount of guaiacyl lignin.

**Keywords:** deep eutectic solvent; bagasse; lignin; cellulose; chemical structure





## 1. Introduction

As a substitute for petrochemical resources, renewable resources have become an important raw material for the green and sustainable development of industries. Ligno-cellulose resources, which are rich in cellulose, hemicellulose, and lignin, are the most widely distributed and abundant renewable resources on earth [1]. Bagasse, a cellulose residue obtained after crushing and pressing sugarcane juice, is an important renewable biomass resource. Sugarcane is an annual crop, and its fiber morphology is closer to that of wood fiber than those of other grass fibers [2]. Therefore, bagasse is considered an ideal fiber material. Bagasse is mainly composed of cellulose, hemicellulose, and lignin, of which cellulose accounts for approximately 46–55% [3]. However, the short fiber, low ratio between length and diameter, and presence of lignin and hemicellulose as fillers and adhesives between cellulose hinder the penetration and diffusion of reagents [4]. A large number of hydroxyl groups on the surface of bagasse leads to strong polarity and water absorption and results in poor interfacial compatibility with most polymers [5]. Thus, the rapid and effective removal of hemicellulose and lignin while maintaining the structure and form of cellulose has been a challenge in the use of bagasse.

In 2001, Abbott et al. prepared a deep eutectic solvent (DES), which is widely used because of its simple preparation process, low toxicity, low price, biodegradability, and good biocompatibility [6]. Choline DES is a kind of solvent formed by covalent and hydrogen bonds between the choline anion and coordination agent [7]. The hydroxyl

group in the solvent can form hydrogen bonds with cellulose, making cellulose relatively stable [8]. Thus, DES shows high solubility toward lignin, and negligible cellulose solubility. Zhang et al. [9] found that DES synthesized by choline chloride with monobasic acid, dibasic acid, and polyol can effectively remove hemicellulose and lignin from corncob. Song et al. [10] extracted 50% lignin from poplar using aqueous gallic acid-based natural deep eutectic solvent. Paula et al. [11] observed guaiacyl units of lignin in a study of two shrub plants, Cistus ladanifer and Erica arborea. Guo et al. [12] extracted 63.4% lignin with typical guaiacyl (G), syringyl (S), and p-hydroxyphenyl (H) structures from corncob with benzyltrimethyl ammonium chloride/lactic acid. Gong Weihua et al. [13] used the acetic acid method to extract lignin from the sunflower seed shell. In their study, the extraction rate of lignin was 70.12%, and the lignin was mainly GS type lignin, in which the content of the guaiac-based structural unit (G) was higher. Liu Liyang et al. [14] studied the pretreatment of peanut straw, peanut shell, and rape straw with ionic liquid, and found that the pretreated samples had few crystal regions and low crystallinity. A study on the low eutectic modification of lignin with choline chloride/glycerol showed that the syrillic group structure (S) of lignin was degraded [15], indicating that DES pretreatment could improve the reactivity of lignin. Therefore, DES was an excellent reagent for the separation of lignocellulose. Acidic DESs were employed in the pretreatment of a lignocellulosic oil palm empty fruit bunch; extraction of higher than 60 wt% of lignin was achieved [16]. A maximum delignification of approximately 90.4% was achieved with lactic acid choline chloride DES at 120 °C for 12 h [17]. When water was added to the lignin-dissolved DES, the solubility of lignin increased by 163–474 times [18]. The lifetime and recyclability of the DES solution showed a recovery yield higher than 90%. More importantly, the pretreatment effect of delignification was largely maintained after recycling [19]. Thus, choline DES can extract lignin from plant fibers with high efficiency and purity, and is an ideal solvent for extracting plant fiber components.

Currently, research on DES mainly focuses on the combination screening of choline chloride and different hydrogen bond donors. Poplar, eucalyptus, and other fiber-rich materials are the primarily used raw materials; there is little research on bagasse. The composition of bagasse materials varies depending on the geographical location and growth environment. The crystallinity of cellulose has a significant influence on the dimensional stability, strength, and heat resistance of the materials [20]. In this study, the fiber morphology, crystallinity, and crystal and chemical structure of bagasse pretreated with five different DESs were examined. The change in the chemical structure of the regenerated lignin was studied to provide a theoretical basis for the processing and application of bagasse biomass.

## 2. Materials and Methods

### 2.1. Material and Reagents

Material: bagasse (obtained from Guangxi province, China) was washed with distilled water at least three times to remove water-soluble impurities. Then, the bagasse was dried in an air-blast drying oven at 60 °C for 24 h. The dried bagasse was comminuted by ultramicro pulverizer (Purchased from Yongkang hongtaiyang electromechanical Co., Ltd, Yongkang, China) and then passed through 60–80 mesh. The screened bagasse was extracted in benzene/ethanol (2:1, *v/v*) solution (100 °C water bath) for 6 h, then placed in an air-blast drying oven at 60 °C for 12 h, and finally stored in a sealed bag for later use.

Reagents: choline chloride, oxalic acid dehydrate, urea, and lactic acid were purchased from Damao Chemical Reagent Factory (Tianjin, China). Glycerol and glycol were purchased from Tianjin Zhiyuan Chemical Reagent Co., Ltd. (Tianjin, China). KBr was purchased from Merck Chemical Technology (Shanghai, China) Co., Ltd. Deuterated methanol was purchased from Aladdin Industrial Co. (Shanghai, China). All other reagents were of analytical grade and used without further purification.

### 2.2. Compositional Analysis of Bagasse

The contents of total cellulose, lignin, ash, and moisture from bagasse were determined according to GB/T2677.10-1995, GB/T10337-1989 and GB/T2677.8-1994, GB/T2677.3-1993, and GB/T2677.2-1993 respectively. The content of cellulose was determined by the nitroethanol method [3,21].

### 2.3. Preparation of DES

The molar ratio of choline chloride to urea (CC-U), ethylene glycol(CC-EG), glycerol (CC-G), and lactic acid (CC-LA) was 1:2, and the molar ratio of choline chloride to oxalic acid (CC-OA) was 1:1, and then mixed at 80 °C until uniform and transparent liquid was formed. Choline chloride, urea, and oxalic acid were placed in a vacuum drying oven for 48 h before use.

### 2.4. Pretreatment of Bagasse Using DES

Bagasse powder (1 g) and DES (20 g) were added into a 50 mL round-bottomed conical flask and placed in an oil bath at 100 °C. Mechanical stirring was performed, and the timer was started when the temperature of the solution reached the 100 °C mark. After 4 h, the conical flask was placed in cold water to stop the reaction. When cooled to room temperature (25 ± 2 °C), 50 mL of absolute ethanol was added to the conical flask, and vacuum filtration was performed using a G2 funnel. The filter cake was washed multiple times with anhydrous ethanol until the filtrate became colorless. The filter cake was dried in a 105 °C oven for 6 h. The excess alcohol was removed by rotary evaporation of the filtrate, the recovered DES can be obtained and 4–5 times volume of distilled water was added to the filtrate. After 48 h of precipitation, the supernatant was removed, and the lignin solid was obtained by centrifugation. The solid was dried in vacuum drying oven at 60 °C for standby.

### 2.5. Materials Characterizations

#### 2.5.1. Amount of Lignin Dissolved in DES

Based on the literature [21], the lignin content of bagasse pretreated with DES was determined. Then, the lignin solubility of bagasse in DES was shown in Equation (1):

$$X\% = \frac{X_1 - X_2}{X_1} \times 100\% \tag{1}$$

where $X_1$ is the content of lignin in bagasse raw material, and $X_2$ is the lignin content in bagasse residue after DES pretreatment.

#### 2.5.2. Scanning Electron Microscopy (SEM)

The morphology of the samples was analyzed using the F16502 scanning electron microscope (Phenom, The Netherlands). All samples were sputter-coated with gold to prevent charging. The accelerating voltage during imaging was 0.5 kV.

#### 2.5.3. X-ray Diffraction (XRD)

XRD patterns of the treated bagasse cellulose were characterized using an X-ray diffractometer (Rigaku D/MAX2500V, Japan Science Corporation, Tokyo, Japan) with Ni-filtered Cu Kα1 radiation at 40 kV and 30 mA at room temperature (25 ± 2 °C). The range of scatter in angle (2θ) was from 5 to 40° at a scan rate of 4 °/min. The crystallinity index (Cr*I*) was calculated by the peak height method based on the diffraction intensity of crystalline and amorphous regions using the following equation [22,23] as shown in Equation (2):

$$Cr I(\%) = \frac{I_2 - I_1}{I_2} \times 100\%, \tag{2}$$

where $I_2$ is the diffraction intensity at $2\theta \approx 22°$, which corresponds to the (020) lattice diffraction and $I_1$ is the minimum intensity at $2\theta \approx 18°$, which corresponds to the amorphous region.

### 2.5.4. Fourier Transform Infrared (FTIR) Spectroscopy

Chemical transformations of bagasse treated under different conditions were recorded on a VERTEX 70 FTIR spectrometer (Bruker, Germany) with samples as KBr pellets. The samples were oven-dried and scanned over the spectral range of 4000–400 cm$^{-1}$ with 32 scans per spectrum.

### 2.5.5. NMR Determination of Lignin Dissolved in DES

Dissolve 10 μL recovered DES in 0.5 mL deuterium methanol and gently shake it to dissolve. The $^1$H spectra were recorded by Bruker DPX-300 NMR spectrometer at 25 °C. The scanning times were 128, the sampling time was 3.17 s, and the relaxation time was 1 s. The methanol solvent peak was used as the internal standard to correct and determine the chemical shifts of other signal peaks.

### 2.5.6. Thermogravimetric Analysis (TGA)

The thermal properties of the samples were analyzed using the TA25 (TA Instruments Corporation, New Castle, DE, USA). Approximately 10 mg of the sample was placed in an Al crucible under a temperature range of 40 to 800 °C and heated at a rate of 10 °C/min in a nitrogen atmosphere.

## 3. Results

### 3.1. Chemical Compositions of Bagasse Fiber

Bagasse was used as the raw lignocellulosic material, its composition is shown in Table 1. The contents of cellulose, hemicellulose, and lignin were 55.07 ± 1.27%, 15.19 ± 0.43%, and 19.84 ± 0.17%, respectively. The contents of acid-soluble lignin and acid-insoluble lignin were 1.57 ± 0.12% and 18.27 ± 0.17%, respectively. The composition obtained in this study varied slightly from that reported in a previous study [3] possibly because of the difference in the geographical location of the raw bagasse.

**Table 1.** Content of bagasse biomass (%).

| Cellulose | Hemicellulose | Lignin | Ash | Moisture | Extract |
|-----------|---------------|--------|-----|----------|---------|
| 55.07 ±1.27 | 25.19 ±0.43 | 19.84 ±0.17 | 0.89 ±0.01 | 8.36 ±0.12 | 1.53 ±0.08 |

### 3.2. Solubility Analysis of Lignin in Different DESs

The lignin solubility after DES pretreatment is shown in Figure 1. The solubility varies significantly with DES formed by different hydrogen bond donors. A maximum solubility of 47.85% is shown by the CC-OA donor, and a minimum solubility of 8.60% is exhibited by the CC-EG donor. The existence of hydrogen bonds in eutectic solvents had an effect on the ether bonds in lignin, which reduced the energy required for their cleavage [8], and the lignin molecules dissolved in the solvent. The solubility of lignin in acidic DES was higher than that in basic DES because the hydrogen bonding is stronger in the former. The hydrogen bonds between the carboxylic acid and chloride ion in the CC-LA and CC-OA solvents were strong. In addition, the carboxyl group (-COOH) can easily react with the hydroxyl group (-OH) on cellulose to form monoesters or cross-linked diesters. Diesters can prevent cellulose from dissolving during acid hydrolysis [24]. CC-LA and CC-OA inhibited the dissolution of cellulose and enhanced the degradation of lignin. In the other three solvents, the hydrogen bond formed between the hydroxyl group and chloride ion was weak; thus, the solubility of lignin was low. However, the solubility of lignin in CC-U

(10.53%) was significantly higher than that reported earlier (2.5%) [25]. This may be because bagasse and wood have different structures.

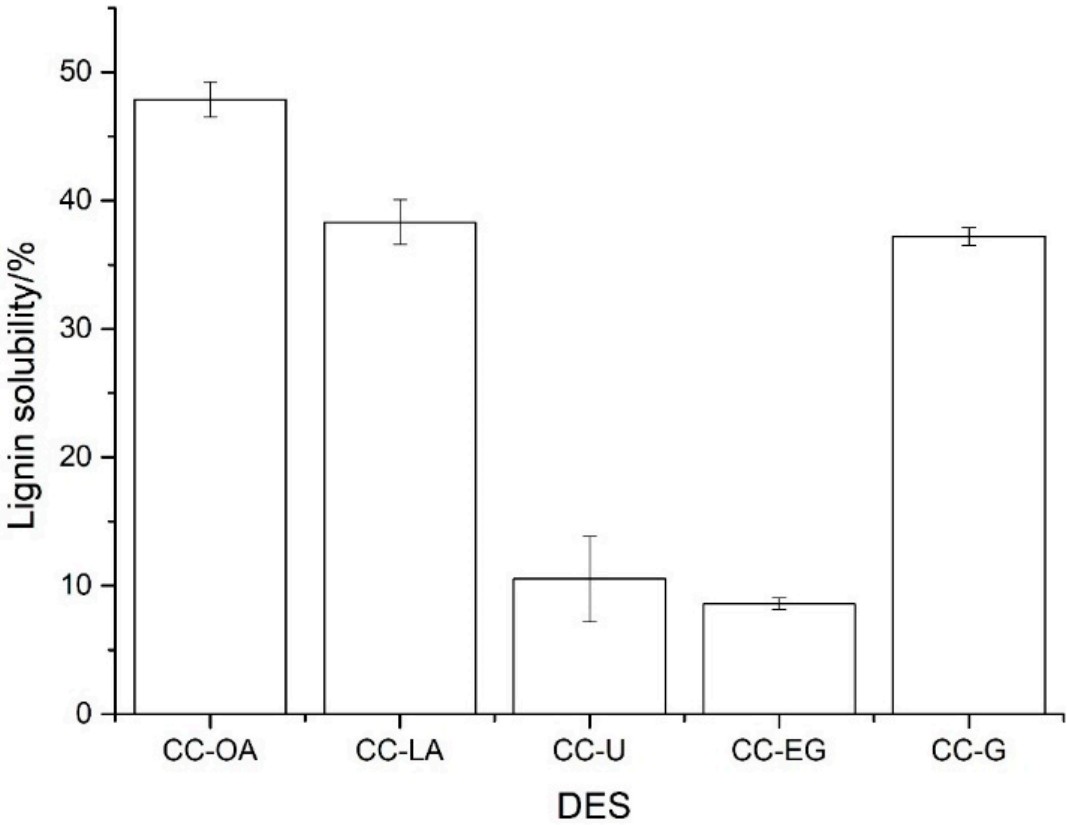

**Figure 1.** Solubility of lignin in deep eutectic solvents (DESs).

### 3.3. Analysis of Surface Morphology of Bagasse

The morphological changes of bagasse fiber under DES pretreatment were observed by SEM. As shown in Figure 2a, cellulose was well-coated with lignin and hemicellulose, and a major portion of the cellulose was unexposed. The bagasse showed long fiber bundles. The rigid connecting structure between lignin and hemicellulose in bagasse was destroyed after DES pretreatment; the fiber bundles became "fragmented," and there were irregular sheet cells, wherein cellulose was exposed. A particularly high amount of cellulose was exposed after the CC-OA treatment. As shown in Figure 2b, the middle lamella of untreated fibers did not fall off completely; the fibers retained a relatively complete and ordered fiber structure, and the surface of the fiber was relatively clean without distinct fuzzing. After DES pretreatment, the intercellular layer of the fibers degraded to different degrees. The fiber surface treated by CC-G and CC-EG exhibited a distinct fuzzing phenomenon. The S1 layer of the fiber was exposed; however, the overall geometric status of the fiber did not change significantly. After pretreatment with CC-U, CC-OA, and CC-LA, the intercellular layer of fiber almost completely fell off, and numerous holes were exposed on the surface of the fiber. As the specific surface area of the fiber increased, the fiber showed different degrees of bending, torsion, and deformation, and the overall geometric status changed significantly. Overall, the surface of the treated fiber exhibited evident peeling off and fine fibrosis.

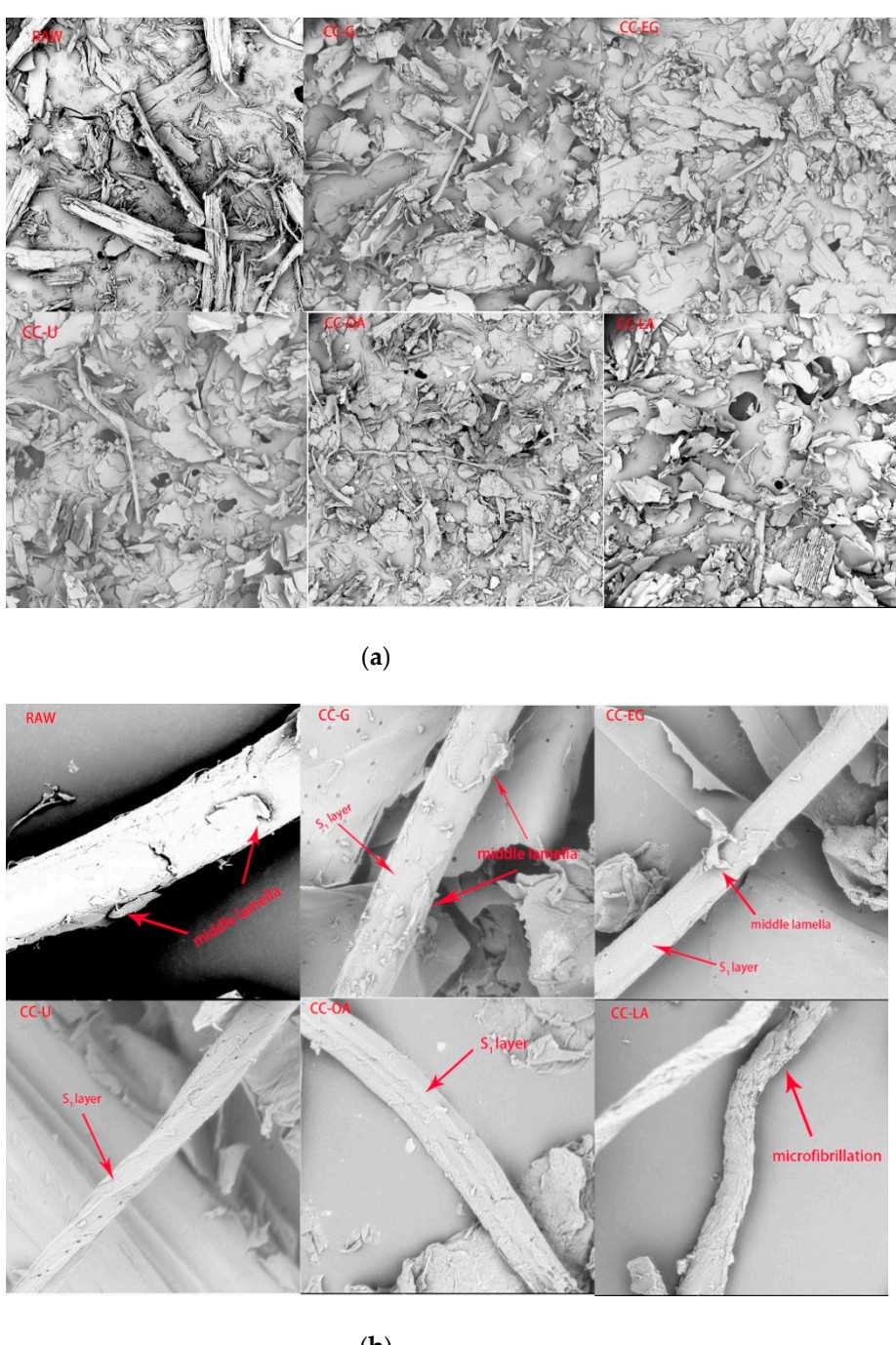

**Figure 2.** SEM images of the original and pretreated bagasse at different magnifications: (**a**) 200×; (**b**) 3000×.

### 3.4. Crystallinity Analysis

The crystal structure and relative crystallinity of the bagasse residue were investigated using XRD (Figure 3). Based on previous studies [26–28], the (110), (110), (200), and (004) crystal planes of cellulose I are generally located at 2θ = 15.0°, 16.4°, 22.5°, and 34.5°. Before and after pretreatment, the samples exhibited a certain absorption at these positions, showing the presence of cellulose I. The (110), (110), and (020) crystal planes of cellulose II are generally located near 2θ = 13.0°, 20.0°, and 22.0°. The major peak in the absorption spectra of the samples was observed at 22.0°; thus, cellulose II was the main crystal form. After CC-U, CC-OA, and CC-LA pretreatment, the crystallinity of bagasse increased. In the CC-OA- and CC-LA-treated samples, the crystallinity increased to 62.26% and 61.65%,

respectively. The increase in crystallinity for the CC-U pretreatment was relatively small. In contrast, after CC-G and CC-EG pretreatment, the crystallinity of bagasse decreased slightly. This may be because of the dissolution of some substances in the crystallization zone during pretreatment [29]. After further analysis, CC-LA was found to be more inclined to separate hemicellulose, and CC-U, CC-G, and CC-EG were more inclined to separate lignin.

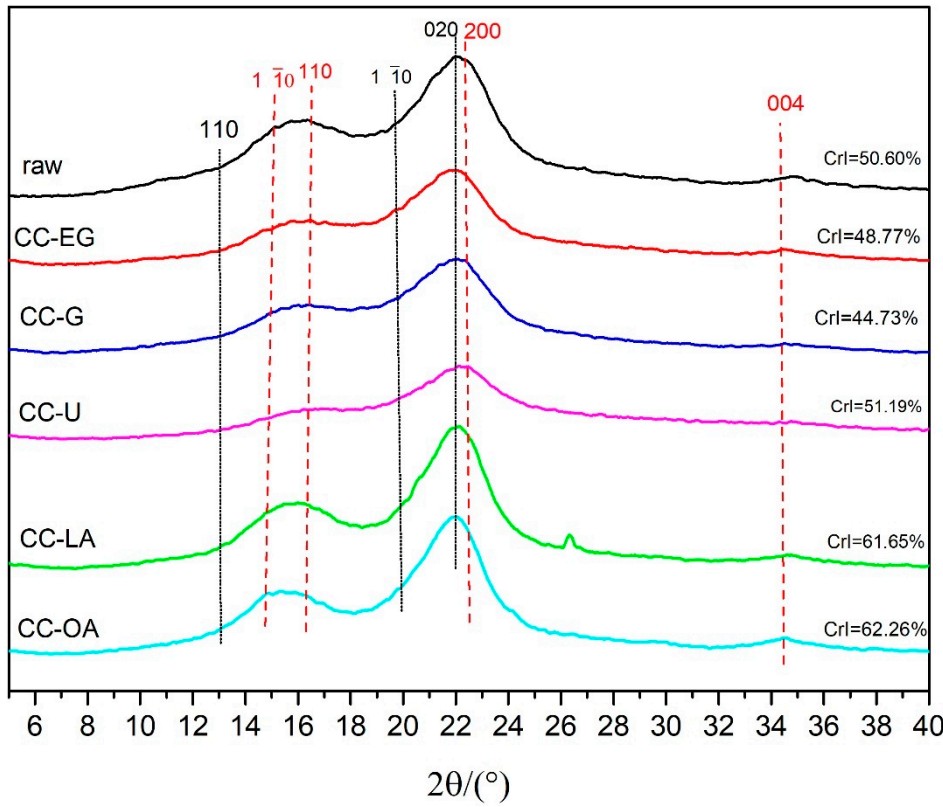

**Figure 3.** XRD patterns of bagasse cellulose treated with different DESs.

### 3.5. Chemical Composition Analysis of Bagasse

Figure 4 shows the FTIR spectra of the original and DES-treated samples. The broad peak at 3300–3500 cm$^{-1}$ in the spectra represents the hydroxyl stretching vibration in lignin or carbohydrate (cellulose or hemicellulose). As the non-fibrous material was removed and more hydroxyl groups were released, the peak moved toward higher frequencies. Generally, the vibration absorption peaks of aromatic ring skeleton are between 1400–1600 cm$^{-1}$. The peaks at 1600, 1510, 1456, and 826 cm$^{-1}$ are the characteristic absorption peaks of lignin. Among them, the peaks at 1600, 1510, and 1456 cm$^{-1}$ are the skeleton vibration absorption peaks of the benzene ring, and this at 826 cm$^{-1}$ is the vibration absorption peaks of C-H linked with the benzene ring [30]. Compared with the untreated bagasse, the absorption peaks of the pretreated samples were weakened, indicating that DES could effectively remove lignin from bagasse. After pretreatment with CC-OA, the peak at 1600 cm$^{-1}$ disappeared, implying that CC-OA pretreatment could remove lignin from bagasse more effectively than the other DES pretreatments. The absorption peak at 1730 cm$^{-1}$ was formed by the vibrations of acetyl groups in hemicellulose/ferulic acid and carboxyl groups in lignin/hemicellulose; the peaks of samples treated with different DESs were weak or absent. This was due to the change in the carboxyl group in hemicellulose, indicating that part of the hemicellulose was lost. After pretreatment with CC-LA and CC-OA, the absorption peak at 1245 cm$^{-1}$ was weakened or absent, implying that the ester bond between lignin and hemicellulose was broken [10]. After DES pretreatment, the vibration absorption peak of the bagasse residue shifted to a higher wavelength at 3400 cm$^{-1}$, indicating that

pretreatment increased the surface hydroxyl groups of cellulose. Based on the FTIR analysis, it was found that DES pretreatment can effectively break the rigid structure between lignin and hemicellulose, such that the bagasse components are separated.

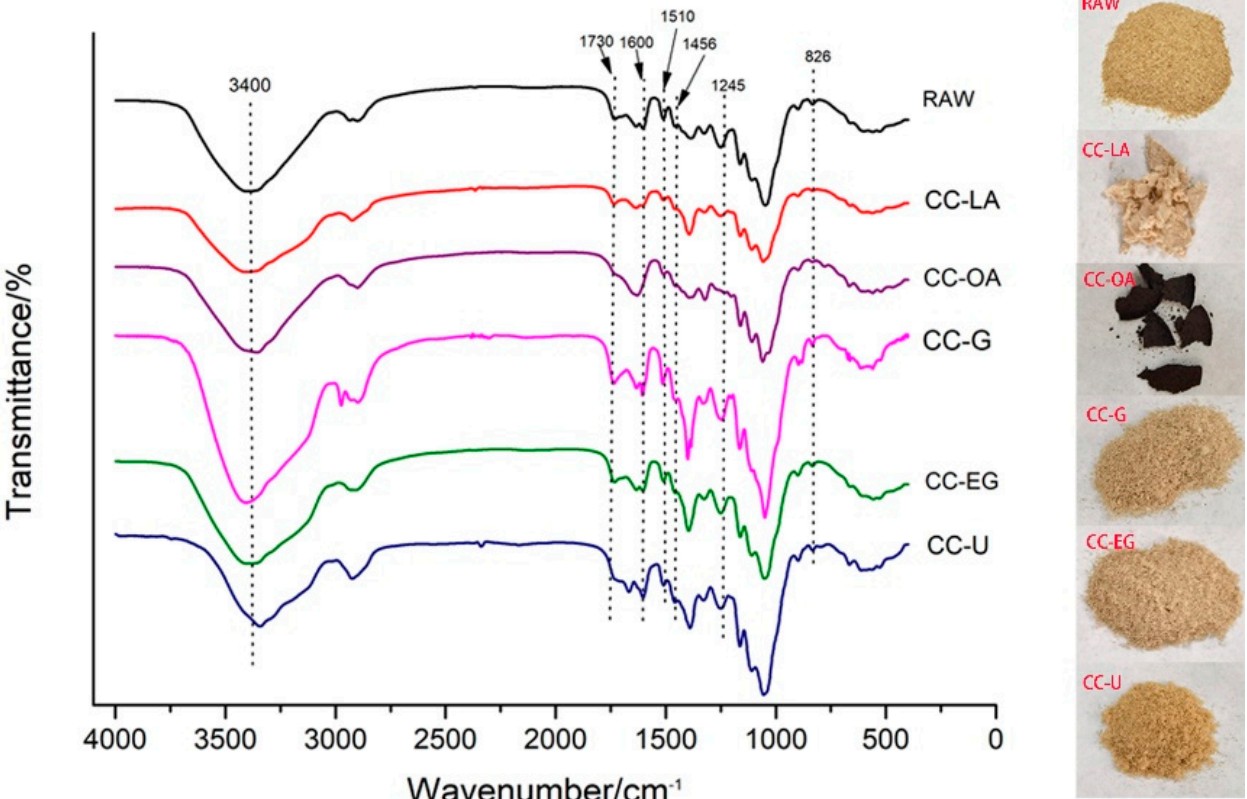

**Figure 4.** FTIR spectra and images of bagasse treated with different DESs.

Based on Figure 4, macro changes in bagasse powder after CC-G, CC-EG, and CC-U treatments are not evident; the dispersed granular form is retained. However, after treatment with CC-LA and CC-OA, the structure is not granular, but flake-like or massive. It was concluded that acidic DES treatment can dissolve lignin more effectively and release a higher number of hydroxyl groups. The presence of hydroxyl groups makes it easier for the original particles to form hydrogen bonds with each other, thus turning the particles into flakes. The carbonization of bagasse after pretreatment with CC-OA can also be observed. Therefore, CC-LA was more suitable for the separation and extraction of bagasse cellulose.

As shown in Figure 5, the stretching vibration of hydroxyl was observed at 3436 cm$^{-1}$. The strong absorption peaks at 1600, 1504, and 1422 cm$^{-1}$ represent the characteristic absorption peaks for the vibrations of the benzene ring skeleton of lignin [31]. The peak at 1463 cm$^{-1}$ is attributed to the C–H deformation combined with aromatic ring vibration, which indicates that the separated part was indeed a lignin component [12]. Thus, the benzene ring skeleton structure was undamaged and well-protected. At approximately 1325 and 1220 cm$^{-1}$, the absorption peaks of the Syringa core and C-H were observed, which indicated that the lignin structural units isolated from bagasse contained several syringyl units (S). The weak peak at 1263 cm$^{-1}$ corresponds to the stretching vibration of C–O in guaiacyl [32]. After treatment with CC-G, CC-U, CC-LA, and CC-OA, the extracted lignin contained a small amount of guaiacyl structural units (G), indicating that the lignin extracted by DES belonged to the G and S types. The absorption peak of the ether bond C-O-C was observed at approximately 1162 cm$^{-1}$. The aryl ether bond between lignin structural units did not break during the DES treatment of bagasse cellulose. When DES

reacted with bagasse components, there was no shear effect on the ether bond of lignin [33]. At 1033 cm$^{-1}$ (corresponding to C-H bending vibration in carbohydrate), the peak of lignin extracted by DES was weak or absent, indicating that the content of carbohydrate was less and the lignin purity was significantly high.

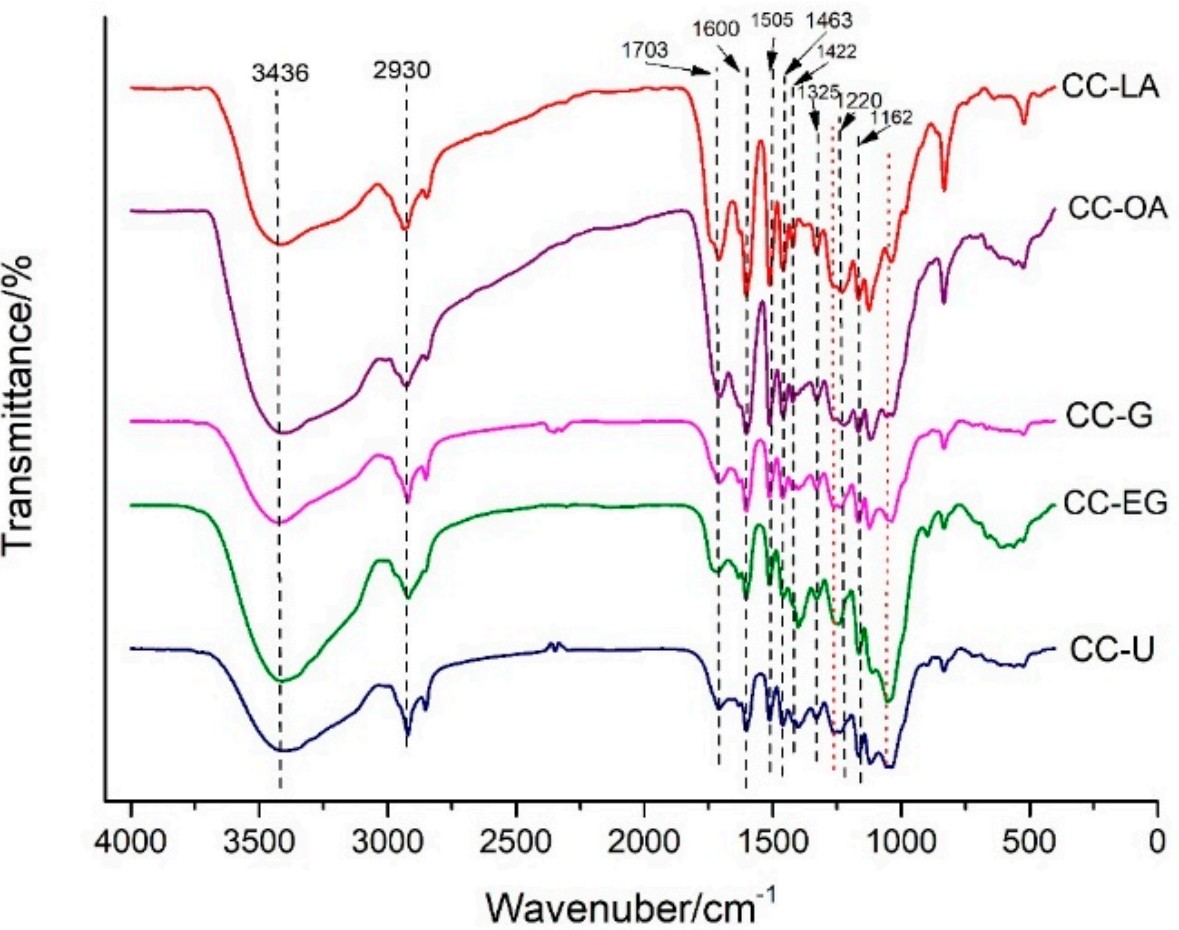

**Figure 5.** FTIR spectra of lignin treated with different DESs.

### 3.6. NMR Analysis of Lignin

According to a previous study, 0.8–1.4 ppm corresponds to the hydrogen content of aliphatic branched chain in lignin [34]. The signals at 3.7 and 2.5 ppm correspond to the protons of methoxy groups in aromatic rings and aliphatic carbon chains, respectively, while 3.3 ppm represents H$_3$ of β-D-xylopyranose. The 4.3 ppm signal corresponds to the γ-acylation of lignin, and the 4.94 ppm signal is in the form of β-β connection, which is the tetrahydrofuran structure [35]. The signal peak of 5.3 ppm corresponds to the hydrogen proton on the α-site carbon of the α-O-4 junction unit [36]. The signal peak of 6.3 ppm corresponds to the hydrogen proton on the α-site carbon of the β-O-4 junction unit [37]. As shown in Figure 6, the lignin dissolved in the five kinds of recovered DESs all had corresponding signal peaks at the position of 0.8–1.4 ppm. Methoxyl protons at 3.7 ppm signal peak can be detected in all samples, the peak signal at 4.3 ppm was detected after CC-LA and CC-OA treatment, which indicated that the activity of DES increased after acid DES pretreatment. The signal peak at 5.3 ppm was only detected in CC-LA and CC-G samples. The signal peaks above 6.0 ppm were very weak or even absent. It shown that the content of lignin and its degradation products was too low, or lignin was decomposed into polysaccharides with smaller molecular weight. This result was similar to that of XiaoJunShen et al. [19].

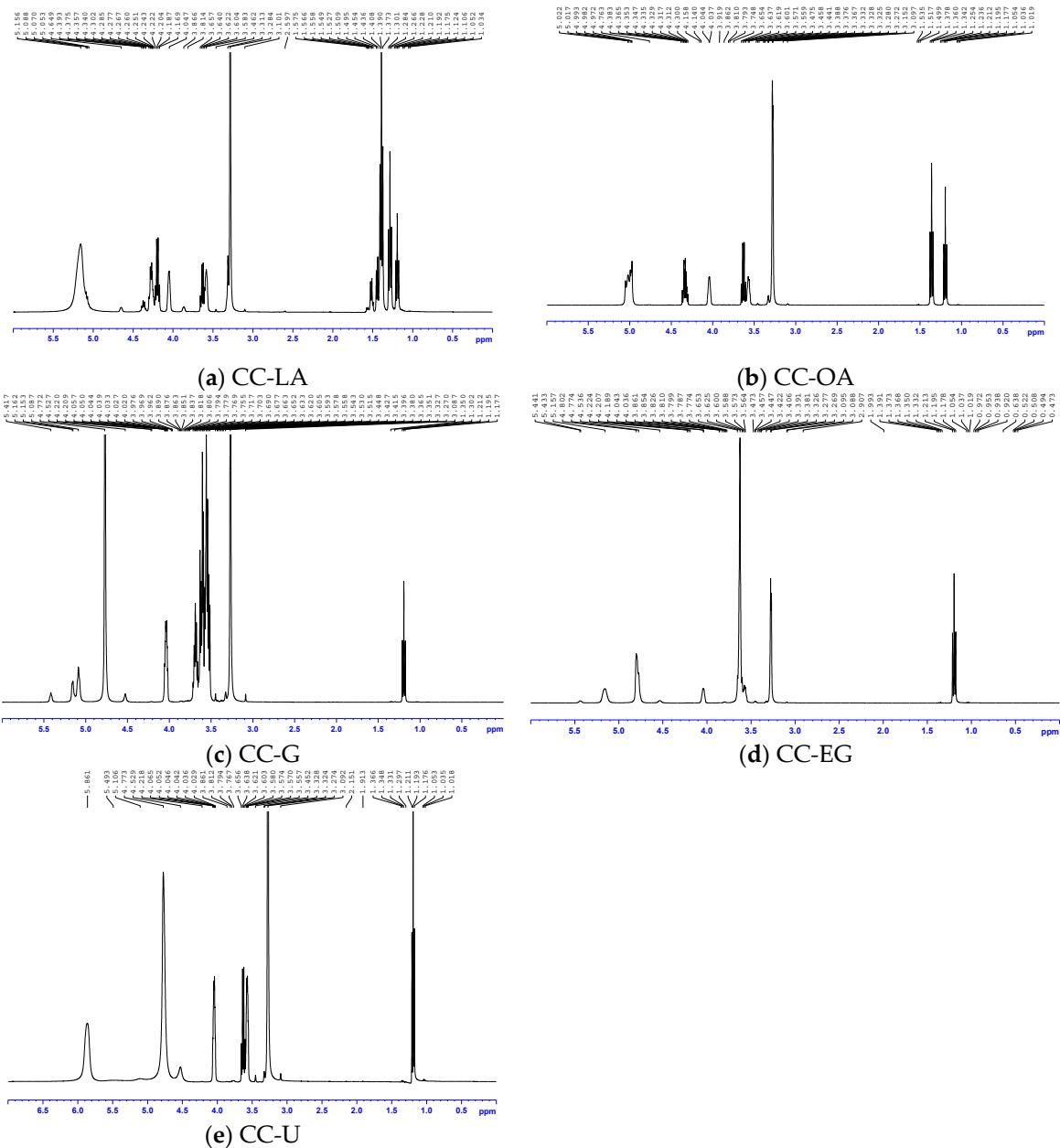

**Figure 6.** 1H-NMR spectra of lignin.

### 3.7. Thermal Stability

According to a previous study, the major thermal degradation stages of hemicellulose, cellulose, and lignin fall in the ranges 275–310 °C, 310–390 °C, and 390–515 °C, respectively [38]. As shown in Figure 7, the samples pretreated by DES have four pyrolysis stages. The first stage occurs at 40–100 °C, and represents the evaporation of water. The second stage occurs at 150–300 °C, and denotes the thermal degradation of hemicellulose. The third stage at 300–400 °C represents the thermal degradation of cellulose; it includes the breaking of the cellulose molecular chains (depolymerization, decomposition, and dehydration of cellulose glycogroups). The fourth stage occurring at 400–550 °C represents the thermal degradation of lignin.

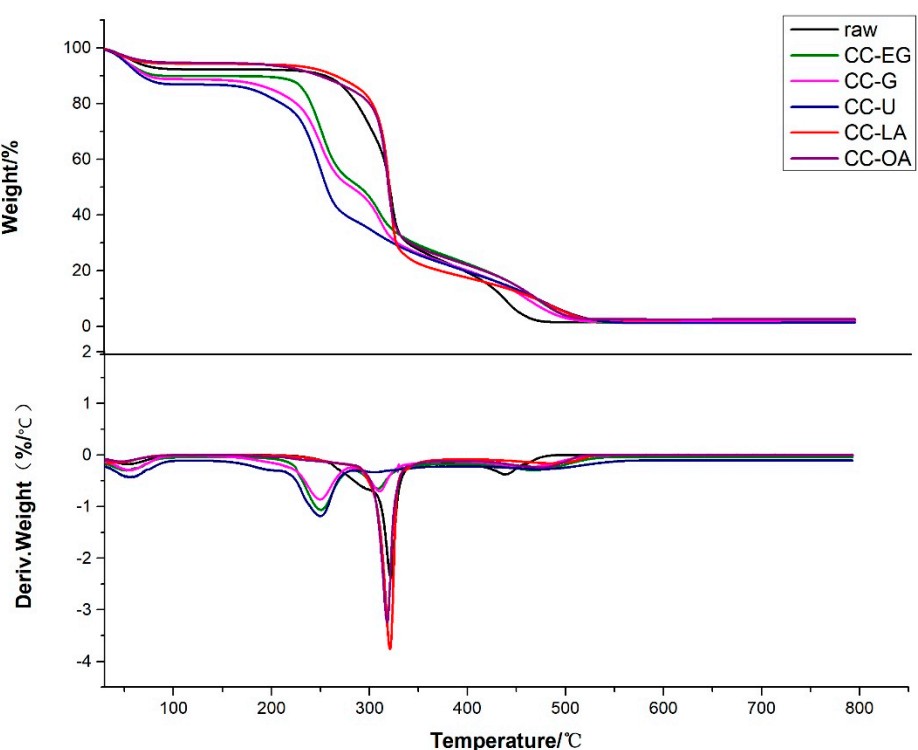

**Figure 7.** TG/DTG curves of bagasse cellulose treated with different DESs.

After CC-OA treatment, the weight loss rate in the second stage of pyrolysis was low (approximately 3.5%), indicating that the hemicellulose had been separated. The highest value of weight loss was approximately 60% at 338 °C, which was the pyrolysis temperature of cellulose. According to a previous study, a higher crystallinity of cellulose leads to better thermal stability [39]. Based on the XRD and SEM analyses, the crystallinity of the pretreated sample was higher, and the exposure rate of cellulose was higher. Therefore, the weight loss rate was mainly concentrated in the pyrolysis stage of cellulose, and the thermal stability was improved. The weight loss rate of CC-OA in the fourth stage was almost zero, indicating that lignin had been successfully separated, which is consistent with the previous FTIR analysis. After CC-LA pretreatment, the weight loss rate of the sample in the second stage of pyrolysis was low, indicating that hemicellulose had been separated from the sample. The pyrolysis range of the third stage was narrow, and the pyrolysis of cellulose in the crystalline region was mainly in this stage. In the fourth stage, the weight loss rate was approximately 22%. A large amount of lignin remained, illustrating that CC-LA mainly caused the separation of hemicellulose, and that of lignin was low. After pretreatment by CC-G, CC-EG, and CC-U, the crystallinity was low and the weight loss rate of pyrolysis was mainly concentrated in the second and third stages, and the fourth stage was significantly small. These results show that these DESs can effectively separate lignin from bagasse and retain most of the hemicellulose and cellulose.

The thermochemical degradation of lignin helps to understand the relationship between its chemical structure and properties. Figure 8 shows the TGA and derivative thermogravimetric (DTG) curves of lignin extracted under different treatment conditions. As shown in Figure 8, the degradation curves of lignin extracted by the five DESs essentially coincide, indicating that the degradation behaviors of the five lignin types were consistent. According to the thermogravimetric behavior of lignin, the lignin degradation can be divided into three stages: initial degradation stage (80–160 °C), main degradation stage (200–470 °C), and carbonization stage (470–600 °C). The initial degradation stage of lignin is the evaporation of water and partial degradation of lignin molecules. The main degradation stage of lignin is the pyrolysis of the lignin network phenylpropane polymer to form small molecular substances [40–42]. During the carbonization stage of

lignin, cleavage of the lignin molecular bond occurred and the residue reacted with coke as the temperature increased; the coke quality was essentially stable after 600 °C.

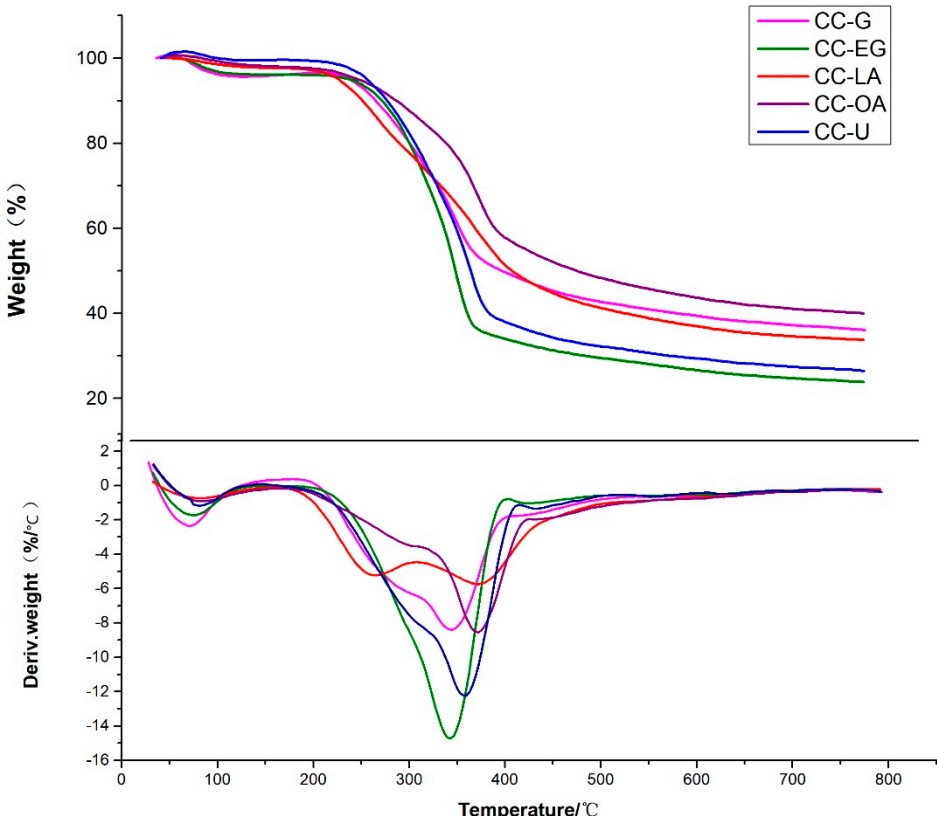

**Figure 8.** TG/DTG curves of lignin treated with different DESs.

## 4. Conclusions

In this study, bagasse biomass was used as a raw material to investigate the effect of hydrogen bond donors on the separation of fiber components and its structure.

The five hydrogen bond donors showed different effects on the separation of bagasse biomass components. The separation effect of the acidic DES was higher than that of the basic DES. The solubility of lignin in CC-OA was the highest (47.85%).

CC-OA can effectively separate lignin and hemicellulose. The lignin separation ability of CC-LA was slightly weaker than that of CC-OA. But after CC-OA pretreatment, bagasse was carbonized, CC-LA was more suitable for the separation and extraction of bagasse cellulose.

CC-G, CC-EG, and CC-U were more inclined to selectively separate lignin than hemicellulose.

After DES pretreatment, the crystalline form of cellulose (mainly cellulose II) was retained. Acidic DES effectively improves the crystallinity of bagasse fiber, but basic DES had little effect on the crystallinity of cellulose.

The lignin dissolved in DES was mainly syringyl lignin; traces of guaiacyl lignin were also dissolved by some DESs.

**Author Contributions:** Data curation and writing-original draft, C.L.; methodology and supervision, C.H., C.Z. investigation, Y.Z. data curation, H.S. software H.Z. conceptualization, S.W. funding acquisition, L.Z., W.L. and L.-J.H. writing-review and editing. All authors have read and agreed to the published version of the manuscript.

**Funding:** The authors are grateful for the financial support from the National Natural Science Foundation of China [215607003] and Guangxi Natural Science Foundation of China (2019JJD120012).

**Data Availability Statement:** The data used to support the findings of this study are available from the corresponding author upon request.

**Acknowledgments:** The authors wish to express their gratitude to the Large Instrument Platform at the Institute of Light Industry and Food Engineering, Guangxi University, China.

**Conflicts of Interest:** The authors declare no conflict of interest.

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
