# Peer review of "Effect of Choline-Based Deep Eutectic Solvent Pretreatment on the Structure of Cellulose and Lignin in Bagasse"

_processes, doi:10.3390/pr9020384_

Round 1
Reviewer 1 Report
The processing of sugarcane and its byproducts is of significant technological interest, and this paper is a useful contribution in this regard. It represents a fairly linear continuation of other work concerning processing of cellulosic plant matter, and so is not groundbreaking, but will be of interest to those working in the field. The work appears to have been competently performed, and consistent with standards for the field.
The English is generally sound, though there are cases where usage is awkward (e.g. in the abstract, "...it has been widely concerned and recognized." There is a need for editing by a native speaker, but this is not a major issue.
There are some problems in the Materials and Methods section, the most egregious of which is that the first 3 paragraphs of the section are text from the template document distributed by the journal. I understand that errors can occasionally creep into even the most meticulously prepared manuscript, but this oversight is so significant that it gives me cause to doubt the merit of the rest of the paper.
Returning to scientific questions, the authors need to state more about the preparation of the bagasse. What volume of benzene alcohol (probably better referred to as "phenol") was used in combination with what weight of cellulose.
It would be helpful if SAXS results were reported as a function q, rather than raw angle, which is an instrument-dependent quantity.
Author Response
Dear editors and reviewer,
Thanks reviewer for these precious comments concerning my manuscript entitled “Effect of choline-based deep eutectic solvent pretreatment on the structure of cellulose and lignin in bagasse(Manuscript ID: processes-1074280)”. These comments are all valuable and very helpful for revising and improving our paper, as well as the important guiding significance to our researches. We are very sorry for the mistakes in this manuscript and the inconvenience they caused in your reading. The manuscript has been thoroughly revised and edited by a native speaker, so we hope it can meet the journal’s standard. Besides, we have studied comments carefully, redone some experiments and have made corrections (marked in red colored words in the manuscript) which we hope to meet with approval. The responses to the reviewer’s comments are as follows:
- Point 1:
- There are some problems in the Materials and Methods section, the most egregious of which is that the first 3 paragraphs of the section are text from the template document distributed by the journal. I understand that errors can occasionally creep into even the most meticulously prepared manuscript, but this oversight is so significant that it gives me cause to doubt the merit of the rest of the paper.
- Returning to scientific questions, the authors need to state more about the preparation of the bagasse. What volume of benzene alcohol (probably better referred to as "phenol") was used in combination with what weight of cellulose.
Response 1: Thanks for your suggestion. I have revised “Material and Reagents”and “Compositional analysis of bagasse”added the explanation (as shown in Line 72 to 77 and).“Material: bagasse(obtained from Guangxi province, China) was washed with distilled water at least three times to remove water-soluble impurities. Then the bagasse was dried in an air-blast drying oven at 60 for 24 hours. The dried bagasse was comminuted by ultramicro pulverizer (Purchased from Yongkang hongtaiyang electromechanical Co., Ltd,China) and then passed through 60-80 mesh. The screened bagasse was extracted in benzene / ethanol (2:1,v/v) solution (100 water bath) for 6h. then placed in an air-blast drying oven at 60 for 12 hours, and finally stored in a sealed bag for later use.
(Line83 to 85) The contents of total cellulose, lignin, ash and moisture from bagasse were determined according to GB /T2677.10-1995, GB / T10337-1989 and GB / T2677.8-1994, GB / T2677.3-1993 and GB / T2677.2-1993 respectively. The content of cellulose was determined by nitroethanol method”
Point 2:It would be helpful if SAXS results were reported as a function q, rather than raw angle, which is an instrument-dependent quantity.
Response 2: Thanks for your kind correction. There are certain differences between XRD and SAXS in principle. XRD is the diffraction phenomenon of X-rays, while SAXS is the diffuse scattering phenomenon of X-rays. The pattern and intensity distribution of SAXS have nothing to do with the atomic composition and crystallization of the scatterer, but only with the shape and size distribution of the scatterer and the density difference between the electron cloud of the surrounding medium. SAXS generally obtains the sample particle size, pore size and other information, and the premise of testing is the density difference inside the sample. XRD can generally obtain crystal plane, crystal plane spacing, crystallinity and other information of the sample. Because the author mainly wants to obtain the crystal plane and crystallinity of the sample after treatment,.Therefore, it is more appropriate to use XRD test here.
And, thanks again for your kind correction. I look forward to hearing from you.
Sincerely,
Chongxing Huang
School of Light Industry & Food Engineering, Guangxi University, 100, Daxue Road, Nanning 530004, China
Tel: +86-138 7818 7372
Email: huangcx21@163.com

Reviewer 2 Report
The manuscript by Li et al. is devoted to the problem of the unstudied effect of different hydrogen bonding donors used in the synthesis of deep eutectic solvents based on choline chloride on the structure and properties of the isolated fractions of lignin and cellulose. The data obtained by the authors provide insight into the mechanisms of interaction of lignocellulosic biomass with deep eutectic solvents during fractionation.
Basically, this work can be interesting for the Processes audience and deserves publication. However, in its present form the manuscript needs improvement.
Specific comments:
- Introduction
1)The literature review contains insufficient data on the crystal and chemical structure of the isolated polysaccharides obtained by other authors with similar solvents.
- Materials and Methods
1) The number of parallel experiments for raw material processing is not specified.
2) The isolated lignin was dried at 105ºC, which may lead to its degradation and oxidation. For correct subsequent characterization of the isolated lignin fractions, vacuum drying at lower temperatures should be used.
3) Molecular weight properties are an important characteristic of the isolated lignin fractions. However, these parameters were not investigated by the authors.
4) One of the criteria for “green solvents” is that they can be reused. However, the paper does not specify information about the reuse (or the possibility of reuse) of solvents in the carried-out experiments.
- Results
1) In the study of lignin solubility, the molecular weight properties of the isolated fractions were not studied. In my opinion, the molecular weight values and the degree of polydispersity of the isolated fractions will promote a better understanding of the reactions occurring during fractionation.
2) No quantitative determination of residual lignin in the treated feedstock was carried out. The presence of lignin will affect the degree of crystallinity of the polysaccharide.
3) IR-spectroscopy does not allow to reach clear conclusions about the changes in the functional composition and the content of interstructural bonds. In my opinion, data obtained by NMR spectroscopy (or other methods) should be added to the lignin characterization.
Some minor specific comments:
1) “Song et al.[10]extracted 50 % lignin from poplar using Aqueous gallic acid-based NADES”. An abbreviation not previously introduced by the authors is used.
2) “Paula et al.[11] observed guaiacyl units in lignin”. The solvent and raw material used are not mentioned.
3) Part of the instructions for authors has not been deleted in Materials and Methods.
4) “VERTEX 70 FTIR spectrometer (Brooklyn, Germany)”. Must be corrected to “Bruker, Germany”
Author Response
Dear editors and reviewer,
Thanks reviewer for these precious comments concerning my manuscript entitled “Effect of choline-based deep eutectic solvent pretreatment on the structure of cellulose and lignin in bagasse(Manuscript ID: processes-1074280)”. These comments are all valuable and very helpful for revising and improving our paper, as well as the important guiding significance to our researches. We are very sorry for the mistakes in this manuscript and the inconvenience they caused in your reading. The manuscript has been thoroughly revised and edited by a native speaker, so we hope it can meet the journal’s standard. Besides, we have studied comments carefully, redone some experiments and have made corrections (marked in red colored words in the manuscript) which we hope to meet with approval. The responses to the reviewer’s comments are as follows:
Point 1:
Introduction
The literature review contains insufficient data on the crystal and chemical structure of the isolated polysaccharides obtained by other authors with similar solvents.
Response 1: Thanks for your suggestion. I have revised “Introduction”and added the explanation (as shown in Line 48 to 54).“Gong Weihua et al. [13] used acetic acid method to extract lignin from sunflower seed shell. In their study, the extraction rate of lignin was 70.12%, and the lignin was mainly GS type lignin, in which the content of guaiac-based structural unit (G) was higher. Liu Liyang et al. [14] studied the pretreatment of peanut straw, peanut shell and rape straw with ionic liquid, and found that the pretreated samples had few crystal regions and low crystallinity. A study on the low eutectic modification of lignin with choline chloride/glycerol showed that the syrillic group structure (S) of lignin was degraded [15], indicating that DES pretreatment could improve the reactivity of lignin.”
Point 2: Materials and Methods
The number of parallel experiments for raw material processing is not specified.
Response 2: Thanks for your kind correction. I have added the specific processing process of the ingredients(as shown in Line 72 to 77). “Material: bagasse(obtained from Guangxi province, China) was washed with distilled water at least three times to remove water-soluble impurities. Then the bagasse was dried in an air-blast drying oven at 60 for 24 hours. The dried bagasse was comminuted by ultramicro pulverizer(Purchased from Yongkang hongtaiyang electromechanical Co., Ltd,China) and then passed through 60-80 mesh. The screened bagasse was extracted in benzene / ethanol (2:1,v/v) solution (100 water bath) for 6h. then placed in an air-blast drying oven at 60 for 12 hours, and finally stored in a sealed bag for later use.”
Point 3:
The isolated lignin was dried at 105ºC, which may lead to its degradation and oxidation. For correct subsequent characterization of the isolated lignin fractions, vacuum drying at lower temperatures should be used.
Response 3: Thanks for your kind correction. I have revised it as your suggestion. And we will deal with these details more rigorously in the future research.
Point 4:
1)Molecular weight properties are an important characteristic of the isolated lignin fractions. However, these parameters were not investigated by the authors.
2)In the study of lignin solubility, the molecular weight properties of the isolated fractions were not studied. In my opinion, the molecular weight values and the degree of polydispersity of the isolated fractions will promote a better understanding of the reactions occurring during fractionation.
Response 4: Thanks for your suggestion. The molecular weight of lignin during dissolution does provide a better understanding of the molecular value and polydispersion of the fraction. To a certain extent, the molecular weight directly affects the solubility of lignin. The author strongly agrees with the editor's opinion and viewpoint. However, in the process of reviewing the literature, the author found that some researchers were studying the solubility of lignin from different raw materials to DES without taking the molecular weight of lignin as a reference value1-4. Based on the fact that this paper is more inclined to the study of structure,the solubility and dispersion of lignin are auxiliary analysis. Therefore, the molecular weight of lignin can not be used as a reference index in this paper
(References:
Li, L., Yu, L., Wu, Z., & Hu, Y. (2019). Delignification of poplar wood with lactic acid-based deep eutectic solvents. Wood Research, 64(3), 507-522.
New, E.K., Wu, T.Y., Lee, C.B.T.L., Poon, Z.Y., Loow, Y.-L., Foo, L.Y.W., et al. (2019). Potential use of pure and diluted choline chloride-based deep eutectic solvent in delignification of oil palm fronds. Process Safety and Environmental Protection, 123, 190-198, doi:10.1016/j.psep.2018.11.015.
Shen, X.-J., Wen, J.-L., Mei, Q.-Q., Chen, X., Sun, D., Yuan, T.-Q., et al. (2019). Facile fractionation of lignocelluloses by biomass-derived deep eutectic solvent (DES) pretreatment for cellulose enzymatic hydrolysis and lignin valorization. Green Chemistry, 21(2), 275-283, doi:10.1039/c8gc03064b.
Tan, Y.T., Ngoh, G.C., & Chua, A.S.M. (2019). Effect of functional groups in acid constituent of deep eutectic solvent for extraction of reactive lignin
Bioresource Technology, 281, 359-366, doi:10.1016/j.biortech.2019.02.010.)
Point 5: One of the criteria for “green solvents” is that they can be reused. However, the paper does not specify information about the reuse (or the possibility of reuse) of solvents in the carried-out experiments
Response 5: Thanks for your suggestion. In the introduction part, some studies on the application of DES to lignin and cellulose were reviewed. Literature [19] mentioned that (as shown in Line 58 to 59)“The lifetime and recyclability of the DES solution showed a recovery yield higher than 90 %. More importantly, the pretreatment effect of delignification was largely maintained after recycling [1].” Recycling involves subsequent research, the authors did not elaborate on it in this article. However, the possibility of recycling is simply mentioned in the introduction, which should meet the standard of green solvent
Point 6:Results No quantitative determination of residual lignin in the treated feedstock was carried out. The presence of lignin will affect the degree of crystallinity of the polysaccharide.
Response 6: Thanks for your suggestion. The researchers measured the content of lignin in bagasse and the solubility of lignin after DES treatment. The residual amount of lignin after pretreatment can be calculated by the following formula:
M=M1-M1*X%
Where M1 is the lignin content in bagasse raw material;X% is the solubility of lignin in different DES.
So the researchers did not measure the lignin residue after pretreatment.The crystallinity of the residue was analyzed by XRD,Therefore, the effect of lignin content in the residue on the crystallinity of polysaccharides was not studied,However we are also very grateful for the valuable advice given by the editor, and we will try our best to take the influence of this part into account in the later research.
Point 7: IR-spectroscopy does not allow to reach clear conclusions about the changes in the functional composition and the content of interstructural bonds. In my opinion, data obtained by NMR spectroscopy (or other methods) should be added to the lignin characterization.
Response 7: Thanks for your kind correction. We have accepted your opinion and added the NMR data(as shown in Line 236 to 248)“According to a previous study [2],0.8-1.4ppm corresponds to the hydrogen content of aliphatic branched chain in lignin [3]. 2.0 ppm corresponds to hydrogen on aliphatic ester carbon and 2.3 ppm corresponds to hydrogen on aromatic ester carbon. The signals at 3.7 ppm and 2.5 ppm correspond to the protons of methoxy groups in aromatic rings and aliphatic carbon chains, respectively. 3 ppm represents H3 of β - D-xylopyranose. The signal peak of 5.3 ppm corresponds to the hydrogen proton on the α - site carbon of the α - O-4 junction unit. The signal peak of 6.3 ppm corresponds to the hydrogen proton on the α - site carbon of the β-O-4 junction unit[2,4-5]. As shown in Fig.6 the lignin dissolved in the five DESs all had corresponding signal peaks at the position of 0.8-1.4ppm. After CC-LA and CC-OA treatment, the peak signal here is relatively stronger and there were more aliphatic branched chains. Methoxyl protons at 3.7ppm signal peak can be detected in all five samples, according to the basic structural units of lignin, the lignin dissolved in the five DESs was GS type. The signal peaks above 6.0 PPM are very weak or even absent. The structure of lignin is not very complicated”
Point 8:“Song et al.[10]extracted 50 % lignin from poplar using Aqueous gallic acid-based NADES”. An abbreviation not previously introduced by the authors is used.
Response 8: Thanks for your comment. I am very sorry for the mistake in this part. I have revised “Line44 to 45,Song et al.[10]extracted 50 % lignin from poplar using aqueous gallic acid-based natural deep eutectic solvent.”
Point 9:“Paula et al.[11] observed guaiacyl units in lignin”. The solvent and raw material used are not mentioned.
Response 9: Thanks for your comment. I am very sorry for the mistake in this part. I have revised “Line45 to 46,Paula et al.[6] observed guaiacyl units of lignin in a study of two shrub plants, Cistus ladanifer and Erica arborea.”
Point 10:“VERTEX 70 FTIR spectrometer (Brooklyn, Germany)”. Must be corrected to “Bruker, Germany”
Response 10:Thanks for your kind correction.We have corrected it
And, thanks again for your kind correction. I look forward to hearing from you.
Sincerely,
Chongxing Huang
School of Light Industry & Food Engineering, Guangxi University, 100, Daxue Road, Nanning 530004, China
Tel: +86-138 7818 7372
Email: huangcx21@163.com

Round 2
Reviewer 2 Report
Point 1. Line 98 - "the recovered des can be obtained" сorrect the abbreviation to "DES", Line 126-127 "Bruker DPX-300 NMR" Point 2. Line 126-127. It is unclear what was investigated - reused solvents, isolated samples, or something else. It is necessary to expand the description of the experiment. The amount of solvent and analyzed sample must be specified. It is necessary to specify the parameters of the spectrum registration or provide a reference to the used method. "1H spectra were analyzed" - In my opinion, it would be more correct to speak not about the analysis, but about the recording of the spectrum. Point 3. Line 238-254. High-molecular-weight lignin fractions are poorly soluble in methanol, and deuterated dimethyl sulfoxide is commonly used for NMR analysis of lignin. Did the entire sample dissolve? Also signals related to the aromatic part of the lignin spectrum usually appear in the region above 6.5 ppm.Thus, the studied samples are probably not lignin, but the products of its depolymerization or its low molecular weight fractions. Therefore, it is necessary to recheck the conclusions drawn. Point 4. Line 252-254. In my opinion, at the moment, the presented images of spectra poorly illustrate the considerations presented by the authors. It is necessary to improve the quality of the images and add explanatory labels on them. Point 5. NMR spectroscopy provides a quantitative or semi-quantitative calculation of the content of the studied functional groups. If it is possible and the authors find it useful, it would be valuable to do this calculation. Point 6. Line 238-247. If the NMR spectra were interpreted according to literature data, it is necessary to provide references for all detected signals or to group them according to the references already provided.
Author Response
Dear editors and reviewer,
Thanks reviewer for these precious comments concerning my manuscript entitled “Effect of choline-based deep eutectic solvent pretreatment on the structure of cellulose and lignin in bagasse(Manuscript ID: processes-1074280)”. These comments are all valuable and very helpful for revising and improving our paper, as well as the important guiding significance to our researches. We are very sorry for the mistakes in this manuscript and the inconvenience they caused in your reading. The manuscript has been thoroughly revised and edited by a native speaker, so we hope it can meet the journal’s standard. Besides, we have studied comments carefully, redone some experiments and have made corrections (marked in red colored words in the manuscript) which we hope to meet with approval. The responses to the reviewer’s comments are as follows:
Point 1.Line 98 - "the recovered des can be obtained" сorrect the abbreviation to "DES", Line 126-127 "Bruker DPX-300 NMR"
Point 2. Line 126-127. It is unclear what was investigated - reused solvents, isolated samples, or something else. It is necessary to expand the description of the experiment. The amount of solvent and analyzed sample must be specified. It is necessary to specify the parameters of the spectrum registration or provide a reference to the used method. "1H spectra were analyzed" - In my opinion, it would be more correct to speak not about the analysis, but about the recording of the spectrum.
Response 1 and 2: Thanks for your suggestion. I have revised and added the specific experimental operation(as shown in Line 126 to 129).“Dissolve 10ul recovered DES in 0.5 mL deuterium methanol and gently shake it to dissolve.the 1H spectra were recorded by Bruker DPX-300 NMR spectrometer at 25 . The scanning times were 128, the sampling time was 3.17s, and the relaxation time was 1 s. The methanol solvent peak was used as the internal standard to correct and determine the chemical shifts of other signal peaks.
Point 3.Line 238-254. High-molecular-weight lignin fractions are poorly soluble in methanol, and deuterated dimethyl sulfoxide is commonly used for NMR analysis of lignin. Did the entire sample dissolve? Also signals related to the aromatic part of the lignin spectrum usually appear in the region above 6.5 ppm.Thus, the studied samples are probably not lignin, but the products of its depolymerization or its low molecular weight fractions. Therefore, it is necessary to recheck the conclusions drawn.
Point 4. Line 252-254. In my opinion, at the moment, the presented images of spectra poorly illustrate the considerations presented by the authors. It is necessary to improve the quality of the images and add explanatory labels on them.
Point 5. NMR spectroscopy provides a quantitative or semi-quantitative calculation of the content of the studied functional groups. If it is possible and the authors find it useful, it would be valuable to do this calculation.
Point 6. Line 238-247. If the NMR spectra were interpreted according to literature data, it is necessary to provide references for all detected signals or to group them according to the references already provided.
Response 3 to 6: Thanks for your kind correction. After receiving your opinion, we once doubted the accuracy of the experiment. We retested a sample and consulted a large number of literatures. It has been proved that the accuracy of our experiment was still acceptable. We re-uploaded the NMR test diagram and changed part of the explanation(as shown in Line 239 to 251): “According to a previous study ,0.8-1.4ppm corresponds to the hydrogen content of aliphatic branched chain in lignin [34]. The signals at 3.7 ppm and 2.5 ppm correspond to the protons of methoxy groups in aromatic rings and aliphatic carbon chains, respectively. 3.3 ppm represents H3 of β - D-xylopyranose. The 4.3ppm signal corresponds to the γ - acylation of lignin, and the 4.94ppm signal is in the form of β - β connection, which is tetrahydrofuran structure[35]. The signal peak of 5.3 ppm corresponds to the hydrogen proton on the α - site carbon of the α - O-4 junction unit[36]. The signal peak of 6.3 ppm corresponds to the hydrogen proton on the α - site carbon of the β-O-4 junction unit[37]. As shown in Fig.6 the lignin dissolved in the five kinds of recovered DESs all had corresponding signal peaks at the position of 0.8-1.4ppm. Methoxyl protons at 3.7ppm signal peak can be detected in all samples, The peak signal at 4.3ppm was detected after CC-LA and CC-OA treatment, which indicated that the activity of DES increased after acid DES pretreatment. The signal peak at 5.3ppm was only detected in CC-LA and CC-G samples.The signal peaks above 6.0 ppm were very weak or even absent. It shown that the content of lignin and its degradation products was too low,or lignin was decomposed into polysaccharides with smaller molecular weight. This result was similar to that of XiaoJunShen et al[19].”
And, thanks again for your kind correction. I look forward to hearing from you.
Sincerely,
Chongxing Huang
School of Light Industry & Food Engineering, Guangxi University, 100, Daxue Road, Nanning 530004, China
Tel: +86-138 7818 7372
Email: huangcx21@163.com